# Inductive or Deductive? Rethinking the Fundamental Reasoning Abilities of LLMs

## Abstract

Reasoning encompasses two typical types: deductive reasoning and inductive reasoning. Despite extensive research into the reasoning capabilities of Large Language Models (LLMs), most studies have failed to rigorously differentiate between inductive and deductive reasoning, leading to a blending of the two. This raises an essential question: *In LLM reasoning, which poses a greater challenge - deductive or inductive reasoning?* While the deductive reasoning capabilities of LLMs, (i.e. their capacity to follow instructions in reasoning tasks), have received considerable attention, their abilities in true inductive reasoning remain largely unexplored due to the inseparability of the two types of reasoning in most of the tasks. To investigate the true inductive reasoning capabilities of LLMs, we propose a novel framework, *SolverLearner*. This framework enables LLMs to learn the underlying function (i.e., $y = f_w(x)$), that maps input data points ($x$) to their corresponding output values ($y$), using only in-context examples. By focusing on inductive reasoning and separating it from LLM-based deductive reasoning, we can isolate and investigate inductive reasoning of LLMs in its pure form via *SolverLearner*. Our observations reveal that LLMs demonstrate remarkable inductive reasoning capabilities through *SolverLearner*, achieving near-perfect performance with ACC of 1 in most cases. Surprisingly, despite their strong inductive reasoning abilities, LLMs tend to relatively lack deductive reasoning capabilities, particularly in tasks involving "counterfactual" reasoning.

## 1 Introduction

Recent years have witnessed notable progress in Natural Language Processing (NLP) with the development of Large Language Models (LLMs) like GPT-3 (Brown et al., 2020) and ChatGPT (OpenAI, 2023). While these models exhibit impressive reasoning abilities across various tasks, they face challenges in certain domains. For example, a recent study (Wu et al., 2023) has shown that while LLMs excel in conventional tasks (e.g., base-10 arithmetic), they often experience a notable decline in accuracy when dealing "counterfactual" reasoning tasks that deviate from the conventional cases seen during pre-training (e.g., base-9 arithmetic). It remains unclear whether they are capable of fundamental reasoning, or just approximate retrieval.

In light of this, our paper seeks to investigate the reasoning capabilities of LLMs. Reasoning can encompasses two types: deductive reasoning and inductive reasoning, as depicted in Fig. 1. Deductive reasoning starts with a general hypothesis and proceeds to derive specific conclusions about individual instances while inductive reasoning involves formulating broad generalizations or principles from a set of instance observations. Despite extensive research into the reasoning capabilities of LLMs, most studies have not clearly differentiated between inductive and deductive reasoning. For instance, arithmetic reasoning task primarily focuses on comprehending and applying mathematical concepts to solve arithmetic problems, aligning more with deductive reasoning. Yet, when employing in-context learning for arithmetic reasoning tasks, where the model is prompted with a few ⟨input, output⟩ examples, the observed improvements are often attributed to their inductive reasoning capacity. This fusion of reasoning types poses a critical question: **Which is the more significant limitation in LLM reasoning, deductive or inductive reasoning?**

To explore this question, it's crucial to differentiate between deductive and inductive reasoning. Current methods that investigate deductive and inductive reasoning often rely on disparate datasets,

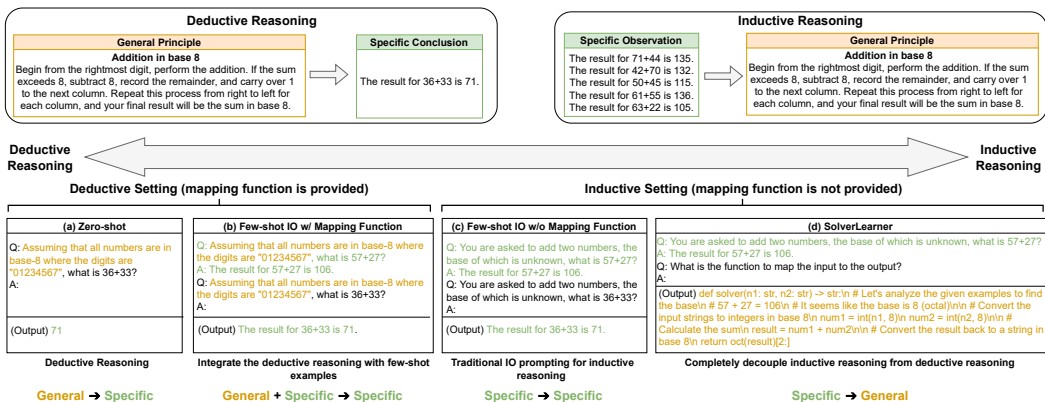

Figure 1: We have designed a set of comparative experiments that utilize a consistent task across different contexts, each emphasizing either *deductive* (i.e., methods (a) and (b)) or *inductive* reasoning (i.e., methods (c) and (d)). As we move from left to right across the figure, the methods gradually transition their primary focus from deductive reasoning to inductive reasoning. Specifically, method (a) is designed to demonstrate the LLMs' *deductive* reasoning in its pure form. Conversely, method (c) utilizes Input-Output (IO) prompting strategies, which are prevalent for probing the *inductive* reasoning skills of LLMs. However, we can observe that methods (c) cannot fully disentangle inductive reasoning from deductive reasoning as their learning process directly moves from observations to specific instances, blurring the lines between the two. To exclusively focus on and examine inductive reasoning, we introduce a novel framework called *SolverLearner*, positioned at the far right of the spectrum.

making direct comparisons challenging (Xu et al., 2023a; Tang et al., 2023; Dalvi et al., 2021; Han et al., 2022; Sinha et al., 2019; Yu et al., 2020). To overcome this limitation, we have designed a set of comparative experiments that utilize a consistent task across different contexts, each emphasizing either deductive (i.e., methods (a) and (b)) or inductive reasoning (i.e., methods (c) and (d)), as depicted in Fig 1. For instance, in an arithmetic task, the proficiency of a LLM in deductive reasoning depends on its ability to apply a given input-output mapping function to solve problems when this function is explicitly provided. Conversely, an LLM's skill in inductive reasoning is measured by its ability to infer these input-output mapping functions (i.e., $y = f_w(x)$), that maps input data points ($x$) to their corresponding output values ($y$), based solely on in-context examples. The base system often serves as the input-output mapping function in an arithmetic task. In line with the aforementioned setup, we employ four methods to investigate the reasoning capacity of LLMs. As we move from left to right across Fig. 1, the methods gradually transition their primary focus from deductive reasoning to inductive reasoning. Method (a), at the far left of the figure, aims to explore the deductive reasoning capabilities of LLMs in its pure form, where no in-context-learning examples are provided (zero-shot settings). While exploring deductive reasoning in its pure form appears relatively straightforward in zero-shot settings, untangling inductive reasoning poses more significant challenges. Recent studies have investigated the inductive reasoning abilities of LLMs (Yang et al., 2022; Gendron et al., 2023; Xu et al., 2023b), they have primarily used Input-Output (IO) prompting (Mirchandani et al., 2023), which involves providing models with a few ⟨input, output⟩ as demonstrations without providing the underlying mapping function. The models are then evaluated based on their ability to handle unseen examples, as illustrated in method (c). These studies often find LLMs facing difficulties with inductive reasoning. Our research suggests that the use of IO prompting might not effectively separate LLMs' deductive reasoning skills from their inductive reasoning abilities. This is because the approach moves directly from observations to specific instances, obscuring the inductive reasoning steps. Consequently, the underperformance in the context of inductive reasoning tasks may be attributed to poor deductive reasoning capabilities, i.e., the ability of LLMs to execute tasks, rather than being solely indicative of their inductive reasoning capability.

To disentangle inductive reasoning from deductive reasoning, we propose a novel model, referred to as *SolverLearner*. Given our primary focus on inductive reasoning, *SolverLearner* follows a two-step process to segregate the learning of input-output mapping functions from the application of these functions for inference. Specifically, functions are applied through external interpreters, such as code interpreters, to avoid incorporating LLM-based deductive reasoning.

We evaluate the performance of several LLMs across various tasks. LLMs consistently demonstrate remarkable inductive reasoning capabilities through *SolverLearner*, achieving near-perfect performance with ACC of 1 in most cases. Surprisingly, despite their strong inductive reasoning abilities, LLMs tend to exhibit weaker deductive capabilities, particularly in terms of "counterfactual" reasoning. This finding, though unexpected, aligns with the previous research. Numerous studies indicate that LLMs lack robust deductive capabilities; for instance, they often struggle with numerical computations. In a zero-shot scenario, the ability of an LLM to correctly execute tasks by applying principles (i.e. deductive reasoning) heavily relies on the frequency with which the model was exposed to the tasks during its pre-training phase (Wu et al., 2023). Conversely, there is significant evidence highlighting the importance of inductive reasoning for LLMs, such as the effectiveness of few-shot in-context learning in enhancing their reasoning abilities. Our study systematically separate the sources of LLM reasoning capabilities and identify the bottlenecks they face. By gaining a more comprehensive understanding of how LLMs reason, we can better identify key areas for improvement. For instance, when designing agent systems, we should leverage the strong inductive capabilities of LLMs while delegating the deductive tasks to external tools.

## 2 Task Definition

Our research is focused on a relatively unexplored question: Which presents a greater challenge to LLMs - deductive reasoning or inductive reasoning? To explore this, we designed a set of comparative experiments that apply a uniform task across various contexts, each emphasizing either deductive or inductive reasoning. The primary distinction between the deductive and inductive settings is whether we explicitly present input-output mappings to the models. Informally, we can describe these mappings as a function $f_w : X \rightarrow Y$, where an input $x \in X$ is transformed into an output $y \in Y$. We distinguish between the deductive and inductive settings as follows:

- **Deductive setting:** we provide the models with direct input-output mappings (i.e., $f_w$).
- **Inductive setting:** we offer the models a few examples (i.e., $(x, y)$ pairs) while intentionally leaving out input-output mappings (i.e., $f_w$).

For example, consider arithmetic tasks, where the base system is the input-output mapping function. The two approaches on the left side of Fig. 1 (i.e., method (a) and (b)) follow the deductive setting, illustrating the case where the arithmetic base is explicitly provided. In contrast, the two methods (i.e., method (c) and (d)) on right side of Fig. 1 adhere to the inductive setting, depicting the scenario characterized by the absence of a specified arithmetic base, while a few input-output examples are provided for guidance.

## 3 Our Framework for Inductive Reasoning: SolverLearner

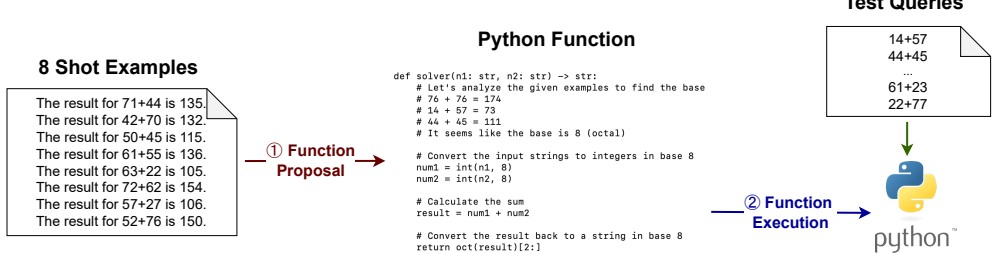

Figure 2: An overview of our framework *SolverLearner* for inductive reasoning. *SolverLearner* follows a two-step process to segregate the learning of input-output mapping functions from the application of these functions for inference. Specifically, functions are applied through external code interpreters, to avoid incorporating LLM-based deductive reasoning.

While recent studies have explored the inductive reasoning abilities of LLMs (Yang et al., 2022; Gendron et al., 2023; Xu et al., 2023b), they have primarily relied on Input-Output (IO) prompt-

ing (Mirchandani et al., 2023). This method involves providing models with a few ⟨input, output⟩ demonstrations and then evaluating their performance on unseen examples, as depicted in method (c) in Fig. 1. Our research suggests that the use of IO prompting and directly evaluating the final instance performance might not effectively separate LLMs' deductive reasoning skills from their inductive reasoning abilities. This is because the approach moves directly from observations to specific instances, obscuring the inductive reasoning steps. To better disentangle inductive reasoning, we propose a novel framework, *SolverLearner*. This framework enables LLMs to learn the function (i.e., $y = f_w(x)$), that maps input data points ($x$) to their corresponding output values ($y$), using only in-context examples. By focusing on inductive reasoning and setting aside LLM-based deductive reasoning, we can isolate and investigate inductive reasoning of LLMs in its pure form via *SolverLearner*. *SolverLearner* includes two-stages as illustrated in Fig. 2:

- **Function Proposal:** In this initial phase, we propose a function, that could be used to map input data points ($x$) to their corresponding output values ($y$). This is corresponding to the inductive reasoning process.

- **Function Execution:** In the second phase, the proposed function is applied through external code interpreters to solve the test queries for evaluation purposes. This phase ensures that the LLM is fully prevented from engaging in deductive reasoning.

### 3.1 FRAMEWORK

In this subsection, we will take the arithmetic task as a case study to demonstrate the entire process.

**Function Proposal:** Given the in-context examples, the primary goal of LLMs is to learn a function that can map input data points ($x$) to their corresponding output values ($y$). This process of learning the mapping between inputs and outputs is akin to inductive reasoning, while employing the learned function to address unseen queries aligns with deductive reasoning. In order to separate inductive reasoning from deductive reasoning, the execution of the learned function should be completely detached from LLMs. To achieve this separation, external tools such as code interpreters serve as efficient way to execute these functions independently. By encapsulating the learned function within Python code, we can effectively detach the duty of deductive reasoning from LLMs, assigning it solely to these external executors. For instance, in function proposal stage for an arithmetic task, we have:

*"You are an expert mathematician and programmer. You are asked to add two numbers, the base of which is unknown. Below are some provided examples: The result for 76+76 is 174.*
*Please identify the underlying pattern to determine the base being used and implement a solver() function to achieve the goal.*
*def solver(n1: str, n2: str) -> str:*
*# Let's write a Python program step by step*
*# Each input is a number represented as a string.*
*# The function computes the sum of these numbers and returns it as a string. "*

**Function Execution:** In the second phase, functions are executed through external code interpreters to solve the test cases for evaluation purposes. These code interpreters act as "oracle" deductive reasoners, fully preventing the LLM from involving deductive reasoning. This ensures that the final results reflect only the inductive reasoning capability of the LLM. To further decouple the LLM's influence in this phase, test cases are generated using a template without involving the LLM. More details can be found in Appendix A.1.3.

## 4 TASKS

In this section, we provide a brief overview of the tasks under consideration. Our focus is on investigating the reasoning abilities of LLMs in both deductive and inductive reasoning scenarios. To ensure a robust evaluation, we carefully select tasks that lend themselves well to comparison. Firstly, to prevent LLMs from reciting tasks seen frequently during pre-training, which could artificially inflate performance in deductive reasoning, a significant portion of the tasks falls into the category of "counterfactual reasoning" tasks. Secondly, in the context of inductive reasoning, where only a few in-context examples are available without the mapping function, our objective is to learn the function that maps inputs to outputs based on this restricted dataset. To achieve this, we choose tasks that are

well-constrained, ensuring the existence of a single, unique function capable of fitting this limited data. Detailed descriptions of each task and the prompts used can be found in Appendix A.1 and A.2.

**Arithmetic** In this study, we focus on the two-digit addition task previously explored in the work of (Wu et al., 2023). We investigate multiple numerical bases, specifically base-8, 9, 10, 11, and 16 where base 10 corresponds to the commonly observed case during pretraining. In the context of deductive reasoning, the base is explicitly provided without any accompanying in-context examples, and the LLM is expected to perform the addition computation by relying on its inherent deductive reasoning abilities. Conversely, in the context of inductive reasoning, instead of explicitly providing the base information to LLMs, we provide LLMs solely with few-shot examples and require them to induce the base through these examples and subsequently generate a function to solve arithmetic problems.

**Basic Syntactic Reasoning** In this setting, we concentrate on tasks related to syntactic recognition previously explored by (Wu et al., 2023). Our objective is to evaluate LLMs using artificially constructed English sentences that vary from the conventional subject-verb-object (SVO) word order. For deductive reasoning, we directly provide the new word order to LLMs without any contextual examples, challenging them to identify the subject, verb, and object within this artificial language. In contrast, for inductive reasoning, we do not give explicit instructions on the changes in word order. Instead, we introduce sentence pairs where one sentence follows the standard word order, and the other follows a modified sequence. Through this setting, LLMs are expected to learn the specific changes made to the word order and then apply this learned rule to identify the subject, verb, and object within new sentences.

**Spatial Reasoning** In this task, we investigate the spatial reasoning previously investigated by (Wu et al., 2023). Our specific focus is on modifying the direction-unit vector mapping and determining the object coordinates in this revised system. We explore multiple systems, starting with the commonly observed case during pretraining, where up corresponds to north, down to south, left to west, and right to east. This is compared to coordinate systems with swapped, rotated, and randomly permuted axes. For deductive reasoning, we directly provide the direction-unit vector mapping without any contextual examples, requiring LLMs to compute the object coordinates within these systems. Conversely, in the context of inductive reasoning, instead of directly explaining the changes made to the direction-unit vector mapping to LLMs, we present LLMs with a few example shots and challenge them to infer the changes made to the mapping. They are then expected to apply this learned function to determine the object coordinates in the system.

**Cipher Decryption** Under this scenario, we explore an innovative task that we have created, concentrating on the decryption of strings encrypted using specific cipher systems. We have incorporated three particular cipher systems for this exploration: the *Alphabetically Sorting Cipher* the *Caesar Cipher* and the *Morse Cipher*. For deductive reasoning, we directly inform LLMs about the cipher system being used, yet we do not offer any contextual examples. The objective for LLMs is to decode strings according to these cipher systems. Conversely, in the inductive reasoning scenario, our task involves providing LLMs with several examples, each consisting of an encrypted string and its decrypted version. The main challenge for the models in this scenario is first to identify what cipher system was used and then to apply that cipher system to decrypt an unseen string.

## 5 RESULTS

For each task, we evaluate our proposed *SolverLearner* for pure LLM inductive reasoning and other settings using two different models, *gpt-3.5-turbo-1106* and *gpt-4-1106-preview*, which are denoted as GPT-3.5 and GPT-4 respectively. Since both methods are closed-source, we do not provide specific information about their size, architecture, and pre-training particulars. To further validate the generalizability of our conclusion, we also included results over additional LLMs, *claude-3-sonnet-20240229-v1:0*, which is denoted as Claude3. Due to space limitations, the full numerical results for Claude3 are provided in Appendix A.4. Our experiments primarily focus on investigating the reasoning abilities of LLMs in both deductive and inductive reasoning scenarios. Therefore, we structure our evaluation across two distinct settings to highlight each type of reasoning. The formal definition of each setting is provided in Sec. 2. To fairly compare inductive and deductive reasoning, in the *deductive setting*, two methods are proposed for investigation:

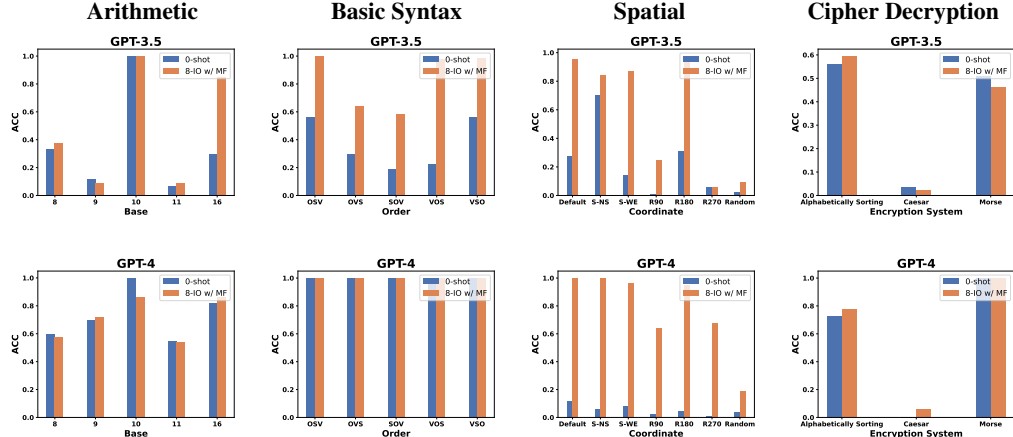

Figure 3: Comparison of the *deductive reasoning abilities* of LLMs across various tasks. Different methods are illustrated through color-coded bars: blue bars indicate the results achieved using Zero-shot, while orange bars show the performance of 8-IO w/ Mapping Function (MF).

- **Zero-shot** evaluates deductive reasoning ability of the LLMs in its pure form. It tests the LLM's ability to conclude information about specific individuals based solely on instructions, without relying on examples.
- **8-IO w/ Mapping Function** (MF) follows the deductive setting but enhances LLM reasoning further by incorporating in-context examples. It aligns with the most commonly used prompt methods for enabling LLM reasoning. With the inclusion of in-context examples, this approach can be seen as leveraging inductive reasoning to augment deductive reasoning.

Similarly, for the *inductive setting*, we also propose two methods for evaluation:

- **8-IO w/o Mapping Function (MF)** aligns with traditional input-output (IO) prompting methods widely used to investigate the inductive reasoning capability of LLMs. However, as this method proceeds directly from a set of observations to specific target instances, it remains intertwined with LLM-based deductive reasoning.
- **8-shot *SolverLearner*** corresponds to our proposed framework for inductive reasoning, capable of evaluating inductive reasoning ability of the LLMs in its pure form. It segregates the learning of input-output mapping functions from the application of these functions for inference, thereby preventing the blend of LLM-based deductive reasoning into the process.

Besides using 8-shot examples, our study also includes experiments with 16-shot examples to assess how changes in the number of in-context examples impact the results. Experimental results are given in the Appendix A.3. Generally, the results indicate that an increase in the number of in-context examples yields only slight improvements across both deductive and inductive reasoning scenarios. Furthermore, we conduct an ablation study concerning our proposed *SolverLearner* in Appendix A.5 for deeper insights into its functionality.

### 5.1 MAIN RESULTS

The results for all tasks are presented from Fig. 3 through Fig. 5. Specifically, Fig. 3 concentrates on comparing performances in the deductive setting, while Fig. 4 examines comparisons in the inductive setting. Additionally, Fig. 5 focuses on contrasting the models' capabilities across deductive and inductive setting. For further reference, the prompts used for all tasks are included in Appendix A.2, and the full numerical results can be found in Appendix A.3.

**LLMs exhibit poor deductive reasoning capabilities, particularly in "counterfactual" tasks.** We include two methods in Fig. 3, Zero-shot and 8-IO w/ Mapping Function (MF), to illustrate the deductive reasoning capability of LLMs. Our observations reveal that LLMs exhibit relatively weaker deductive capabilities, especially in "counterfactual" tasks, while showing prowers in standard

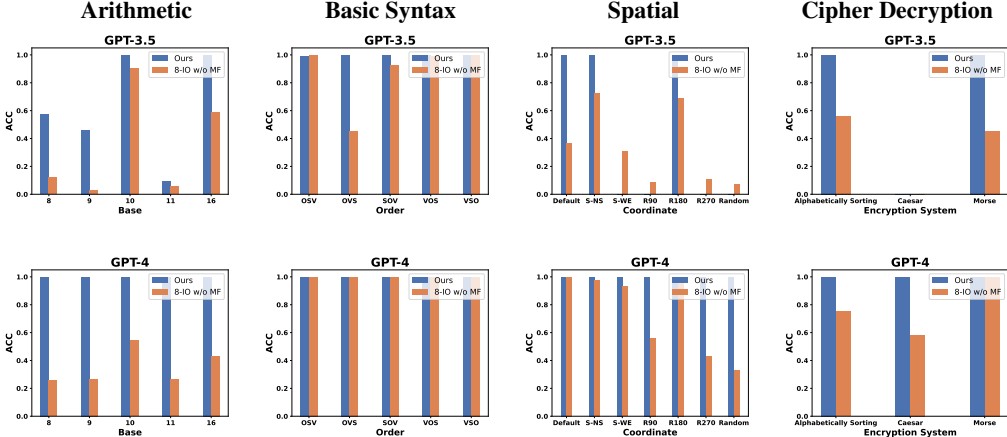

Figure 4: Comparison of the *inductive reasoning abilities* of LLMs across various tasks. Different methods are illustrated through color-coded bars: blue bars indicate the results achieved using our proposed *SolverLearner*, while orange bars show the performance of 8-IO w/o Mapping Function (MF).

tasks like base-10 arithmetic. This aligns with findings reported in (Wu et al., 2023). Integration of in-context examples notably enhances LLMs' performance in various scenarios, suggesting that their improvement stems from the acquisition of knowledge through inductive reasoning from these examples. This further confirms the exceptional inductive reasoning abilities of LLMs. This combined evidence suggests that LLMs face challenges in precisely following instructions and executing commands, especially when those instructions are relate to scenarios rarely encountered during their pre-training phase.

**LLMs demonstrate remarkable inductive reasoning capabilities through *SolverLearner*.** We include two methods in Fig. 4, *SolverLearner (Ours)* and 8-IO w/o Mapping Function (MF), to illustrate the inductive reasoning capability of LLMs. While 8-IO w/o Mapping Function (MF) struggles with inductive reasoning, *SolverLearner* consistently achieves perfect performance with an accuracy of 1 across all the cases with GPT-4 and succeeds in most cases when used with GPT-3.5. This discrepancy arises because the utilization of IO prompting to directly reach conclusions on target instances may not effectively distinguish between LLMs' deductive and inductive reasoning skills. By completely disentangling the inductive reasoning of LLMs, our proposed *SolverLearner* shows the remarkable inductive reasoning capabilities inherent in LLMs. It is also noteworthy that the efficacy of LLMs' inductive reasoning capability heavily depends on the foundational model, with GPT-4 consistently outperforming GPT-3.5.

**Deductive reasoning presents a greater challenge than inductive reasoning for LLMs.** To compare the challenges of the deductive reasoning capability with the inductive reasoning capability of LLMs, we include two methods in Fig. 1, *SolverLearner* and Zero-shot, demonstrating pure inductive and deductive reasoning abilities. Since the entire reasoning involves two steps: first, obtaining the input-output function ($f_w$), which corresponds to inductive reasoning, and second, applying the function for inference, which corresponds to deductive reasoning. Once both steps are successfully completed, perfect performance is observed, as indicated by the dotted line in the figure. Zero-shot can be seen as replacing the first step with an oracle, with deductive reasoning capability of LLMs to be studied, while *SolverLearner* can be seen as replacing the second step with an oracle, with inductive reasoning capability of LLMs to be studied. By comparing the gaps of *SolverLearner* and Zero-shot towards perfect reasoning, we can observe that in most cases, LLMs can complete the inductive step perfectly, while they rarely achieve perfect performance on the deductive step. This indicates that in LLM reasoning, deductive reasoning presents a greater challenge. Note that we avoid to phrasing it as directly comparing inductive and deductive reasoning capabilities. Instead, we examine whether the gaps mainly come from inductive or inductive reasoning, considering that LLMs could not achieve perfect counterfactual reasoning.

## 5.2 Ablation Study

We conducted several experiments to gain a deeper understanding of our framework, detailed in the ablation studies in Appendix A.5. These experiments include investigating the effects of programs executed by a Python interpreter v.s. natural language executed by an LLM and examining the impact of the number of in-context learning examples.

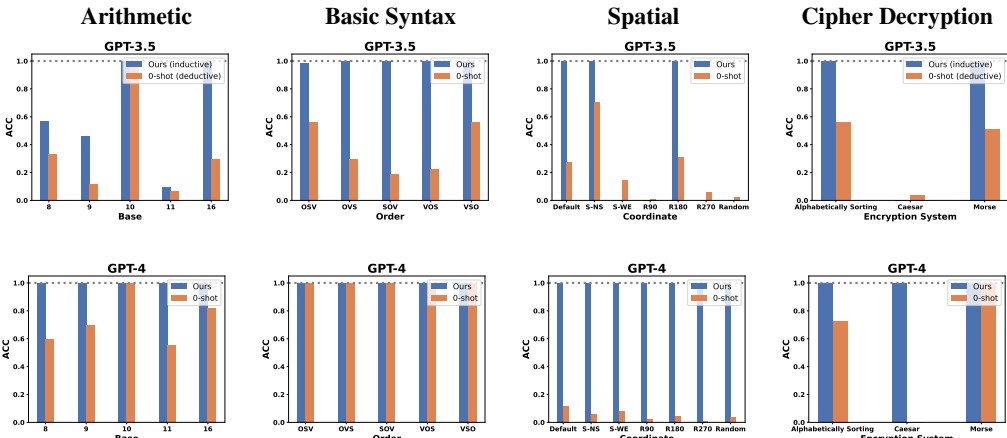

Figure 5: Comparison of the *inductive reasoning abilities versus deductive reasoning abilities* of LLMs across various tasks. Different methods are illustrated through color-coded bars: blue bars indicate the results achieved using our proposed *SolverLearner* for *inductive reasoning*, while orange bars show the performance of Zero-shot for *deductive reasoning*.

## 6 Related Works

### 6.1 In-Context Learning

GPT-3 (Brown et al., 2020) has demonstrated its effectiveness in learning from a few demonstration examples and solve previously unseen tasks without requiring updates to its model parameters (Wei et al., 2022a). This remarkable capability is commonly referred to as the "in-context learning ability" of language models. It implies that the LLMs can leverage its existing knowledge and generalize from a few demonstration examples to solve new, related tasks (Dong et al., 2022; Liu et al., 2021; Rubin et al., 2021; Gonen et al., 2022). Some notable works include chain-of-thought (CoT) prompting (Wei et al., 2022b), which elicits reasoning with intermediate steps in few-shot exemplars. Built upon the CoT framework, several works expand CoT by organizing and processing thoughts using more complex structures, such as trees (Yao et al., 2023) and graphs (Besta et al., 2023) or breaking a problem into sub problems and then proceeds to solve each one independently (Zhou et al., 2022). While these studies have effectively improved the reasoning capability of LLMs, they have failed to clearly distinguish between inductive and deductive reasoning, let alone investigate which represents a more critical limitation for LLM reasoning capabilities: deductive reasoning or inductive reasoning.

### 6.2 Exploring LLMs' Reasoning Skills

Despite the impressive achievements of LLMs in various reasoning tasks, the underlying mechanisms of their reasoning capabilities remain a subject of debate. The question of whether LLMs genuinely reason in a manner akin to human cognitive processes or merely simulate aspects of reasoning without true comprehension is still open (Huang & Chang, 2022). For instance, Kojima et al. have suggested that LLMs exhibit commendable zero-shot reasoning abilities, implying that these models can draw logical conclusions in scenarios they have not been explicitly trained on (Kojima et al., 2022). However, some researchers cast doubt on the reasoning capability of LLMs. While approaches like the chain-of-thought method may mimic human-like thought processes, it remains uncertain whether LLMs are genuinely engaging in reasoning or simply following patterns learned during training (Wei et al., 2022b; Valmeekam et al., 2022). Additionally, there's a debate regarding whether LLMs are

symbolic reasoners (Tang et al., 2023) or possess strong abstract reasoning capabilities (Gendron et al., 2023). In light of these seemingly contradictory conclusions, our research aims to investigate deeper into the reasoning capabilities of LLMs. We intend to dissect the nuances of inductive and deductive reasoning within the context of LLMs, identifying which form of reasoning presents a more significant challenge to their reasoning abilities.

### 6.3 Equipping LLMs with External Tools

Large Language Models (LLMs) have made significant progress in utilizing tools through frameworks like Program of Thoughts (PoT) (Chen et al., 2022), CREATOR (Qian et al., 2023) and LATM (Cai et al., 2023), which allow LLMs to create tools using documentation and code. Logic-LM (Pan et al., 2023) integrates LLMs with symbolic solvers to improve logical problem-solving, However, these approaches focus exclusively on deductive reasoning, aiming to enable LLMs to derive correct answers for specific questions without incorporating the capacity for inductive reasoning to infer underlying mapping function shared by few-shot examples. In contrast, our primary objective is not to propose a new framework for using tools to enhance the problem-solving capabilities of LLMs. Instead, we aim to differentiate between deductive and inductive reasoning within LLMs and explore which presents a greater challenge to their reasoning abilities.

## 7 Limitations

**LLMs cannot perform inductive reasoning over all the tasks** In our inductive learning setting, LLMs are provided with only a limited number of contextual examples. The goal is to infer the function that accurately maps inputs to outputs based solely on this constrained dataset. In order to solve this problem, it is significant that we can find a unique function satisfied given these examples. For instance, a linear function can be precisely determined given just two data points, as it has a singular solution. However, attempting to deduce a quadratic curve from two points poses an insurmountable challenge due to the existence of infinite functions capable of passing through those specific points. Additionally, LLMs might struggle to discern the correct mapping function when the search space of the problem expands excessively. Consider the case of arithmetic tasks; without limiting the search space to finding a suitable base that aligns with the observations, the task becomes overwhelmingly complex. This is because the search space could encompass any conceivable rule that accommodates the observations.

**The effectiveness of LLMs' inductive reasoning capability is heavily reliant on the foundational model** While GPT-4 consistently showcase impressive inductive reasoning abilities through *Solver-Learner* and achieve perfect performance with ACC of 1 across all the tasks, GPT-3.5 struggle to learn the correct input-output mapping function in several cases. This observation suggests that the inductive reasoning potential of LLMs is significantly constrained by the underlying model.

**Chain of Thought (COT) has not been incorporated into the comparison** Chain of Thought (COT) (Wei et al., 2022b) is a significant prompting technique designed for use with LLMs. Rather than providing a direct answer, COT elicits reasoning with intermediate steps in few-shot exemplars. This method was not incorporated into our comparison as it is viewed as a technique to improve the deductive reasoning capabilities of LLMs. Although COT has proven to be effective across various tasks, numerous studies highlight a significant performance gap that COT still needs to bridge to achieve flawless execution.

## 8 Conclusion

This study aims to explore a less-investigated aspect of LLMs: within LLM reasoning, which presents a greater challenge — deductive or inductive reasoning? To investigate the inductive reasoning capacities of LLMs, we introduce a novel framework called *SolverLearner*. By concentrating on inductive reasoning while setting aside LLM-based deductive reasoning, *SolverLearner* can scrutinize the pure form of inductive reasoning in LLMs. Our findings unveil remarkable inductive reasoning prowers in LLMs through *SolverLearner*, achieving near-perfect performance with an ACC of 1 in most cases. Surprisingly, despite their strong inductive reasoning abilities, LLMs often exhibit weaker deductive capabilities, particularly in tasks involving "counterfactual" scenarios.

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

## A  APPENDIX

### A.1  FULL SETUPS

SolverLearner is a prompting based reasoning approach, and we only need to perform inference with LLMs.

#### A.1.1  SETTINGS FOR EACH TASK

**Arithmetic** The arithmetic dataset introduced in Wu et al.'s paper (Wu et al., 2023) comprises 1,000 randomly selected addition expressions, each involving two-digit numbers. These expressions are drawn from bases 8, 9, 10, 11, and 16, with separate sampling for each base. Importantly, all the expressions have been carefully chosen to yield distinct results when evaluated in their respective bases, thereby distinguishing them from one another during the process of rule learning.

**Basic Syntactic Reasoning** In accordance with the methodology outlined in Wu et al.'s work (Wu et al., 2023), we have generated a set of 100 simple three-word sentences (e.g., "bob likes bananas") with five different word order variations (e.g., "bananas bob likes" in OSV format). Subsequently, we tasked LLMs with learning how to manipulate sentence order. It's noteworthy that we took great care in selecting words to ensure that each word in a sentence can only fulfill one specific role, such as subject, object, or verb. For instance, we ensured that sentences like "bob likes anna" were excluded, as both "bob" and "anna" could potentially serve as both subjects and objects, violating this constraint.

**Spatial Reasoning** The spatial reasoning dataset introduced in Wu et al.'s paper (Wu et al., 2023) consists of 100 rooms that were randomly selected, and each room contains three distinct objects. The spatial directions within these rooms are represented using unit vectors. For instance, north is represented as (0, 1), south as (0, -1), east as (1, 0), and west as (-1, 0), with a y-axis pointing upward serving as the default orientation. In our study, we have modified the mapping between directions and unit vectors and tasked LLMs with learning this new direction-to-unit vector relationship. We explore two direction-swapped scenarios (north-south and east-west), three rotated scenarios (by 90°, 180°, and 270°), and a randomly permuted scenario. The primary metric we report is instance-level accuracy, which necessitates that all three objects within a room must be correctly positioned in order to be considered accurate.

**Cipher Decryption** We've generated a collection of 100 pairs of strings (e.g., "Mrxuqhb -> Journey" for Caesar Cipher) for each of three cipher systems, including the *Alphabetically Sorting Cipher* the *Caesar Cipher* and the *Morse Cipher*. Each pair comprises an encrypted string (e.g., "Mrxuqhb") and its corresponding decrypted version (e.g., "Journey"). By providing LLMs with several examples, each containing an encrypted string alongside its corresponding decrypted counterpart, the primary task is to accurately determine the cipher system employed in an open-world context.

#### A.1.2  FEW SHOT EXAMPLE GENERATION

The preparation of examples for few-shot learning follows a straightforward process. We divide all the data into a training set and a test set, from which few-shot examples are extracted from the training set. These few-shot examples are automatically prepared by associating queries with their corresponding ground truth answers using a pre-defined template.

#### A.1.3  TEST CASE GENERATION

In the function execution phase, the test cases are generated using a template without involving LLM. In particular, the test cases are drawn from the test data files, containing all the queries along with their correct answers (e.g., "76+76 = 174"). When the LLM is used for generating code, we specify a function interface, such as def solver(n1: str, n2: str) -> str. Then, using the query examples provided, like "76+76 = 174", we create test cases by applying this function interface to the query (e.g., solver(76,76)), thereby eliminating any reliance on LLM for this process. This method ensures that our test case generation is 100% correct.

Table 1: Prompts for the Arithmetic Task.

| Mode | Prompt |
|---|---|
| Zero-shot | You are a mathematician. Assuming that all numbers are in base-8 where the digits are "01234567", what is 36+33? End the response with the result in "\boxed{result}". |
| Few-shot IO w/ MF | You are a mathematician. You are asked to add two numbers. Assuming that all numbers are in base-8 where the digits are "01234567". Below are some provided examples:
The result for 76+76 is 174.
Please identify the base being used and determine what is 36+33? End the response with the result in "\boxed{result}". |
| Few-shot IO w/o MF | You are a mathematician. You are asked to add two numbers, the base of which is unknown. Below are some provided examples:
The result for 76+76 is 174.
Please identify the base being used and determine what is 36+33? End the response with the result in "\boxed{result}". |
| SolverLearner | You are an expert mathematician and programmer. You are asked to add two numbers, the base of which is unknown. Below are some provided examples:
The result for 76+76 is 174.
Please identify the underlying pattern to determine the base being used and implement a solver() function to achieve the goal.
def solver(n1: str, n2: str) -> str:
# Let's write a Python program step by step
# Each input is a number represented as a string.
# The function computes the sum of these numbers and returns it as a string.
After defining the solver() function, create test cases based on the input examples and print the results. An example of a test case could be "print(solver("76", "76"))". Place the function solver() as well as the test cases between "START_CODE" and "END_CODE". |

## A.2 FULL PROMPTS

We provide the prompts that we used to query the LLMs for all tasks in Tables 1 to 4. We do not use the system message field for any model.

## A.3 FULL RESULTS

We show the full numerical results in Tables 5 to 8. In addition to using 8-shot examples, these results also include experiments with 16-shot examples to assess how changes in the number of in-context examples impact the results.

## A.4 MORE RESULTS ON ADDITIONAL LLMS

To validate the generalizability of our conclusion, we have included additional LLMs, *claude-3-sonnet-20240229-v1:0*, which is denoted as Claude3. We show the full numerical results in Tables 9 to 12.

## A.5 ABLATION STUDIES

**LLMs struggle as executors when applying learned functions.** To better demonstrate the deductive capacity of LLM, we present both GPT-3.5 and Python with identical code and task them with applying the code to deduce the same set of queries. As shown in Table 13, while the Python interpreter can be considered an oracle, delivering flawless performance, it proves challenging for LLMs to accurately execute the code.

**LLMs can learn the function with very few examples when the inductive reasoning problem is well defined.** To examine the impact of the number of few-shot examples on the inductive reasoning capability of LLMs, we vary the number of in-context examples within [1,2,4,8,16] and assess performance on the spatial reasoning task using GPT-3.5 as presented in Table 14. We observe that even with very few examples, GPT-3.5 can still learn the mapping function if it is learnable.

Table 2: Prompts for the Basic Syntactic Reasoning Task.

| Mode | Prompt |
|---|---|
| Zero-shot | You are an expert in linguistics. Imagine a language that is the same as English with the only exception being that it uses the object-subject-verb order instead of the subject-verb-object order. Please identity the subject, verb, and object in the following sentences from this invented language:
shirts sue hates.
Encode the identified subject, verb, and object in the form of a dictionary with the following structure: {'subject': ?, 'verb': ?, 'object': ?}. |
| Few-shot IO w/ MF | As a linguistics expert, your objective is to analyze sentences in a constructed language that shares English vocabulary but uses the object-subject-verb order instead of the subject-verb-object order. Presented below are examples of valid sentences in this constructed language, accompanied by their corresponding English translations.
A sentence in this invented language: phones mary finds. Its equivalent sentence in English reads: mary finds phones.
Following the examples, please analyze the subject, verb, and object in the following sentences from this invented language:
shirts sue hates.
Encode the identified subject, verb, and object in the form of a dictionary with the following structure: {'subject': ?, 'verb': ?, 'object': ?}. |
| Few-shot IO w/o MF | As a linguistics expert, your objective is to analyze sentences in a constructed language that shares English vocabulary but follows a unique grammatical structure. Presented below are examples of valid sentences in this constructed language, accompanied by their corresponding English translations.
A sentence in this invented language: phones mary finds. Its equivalent sentence in English reads: mary finds phones.
Following the examples, please analyze the subject, verb, and object in the following sentences from this invented language:
shirts sue hates.
Encode the identified subject, verb, and object in the form of a dictionary with the following structure: {'subject': ?, 'verb': ?, 'object': ?}. |
| SolverLearner | As a linguistics expert, your objective is to analyze sentences in a constructed language that shares English vocabulary but follows a unique grammatical structure.Presented below are examples of valid sentences in this constructed language, accompanied by their corresponding English translations.
A sentence in this invented language: phones mary finds. Its equivalent sentence in English reads: mary finds phones.
Please summarize the pattern concerning the order of subject, verb and object in this invented linguistic system. Place the pattern between START_PATTERN and END_PATTERN. |

Table 3: Prompts for the Spatial Reasoning Task.

| Mode | Prompt |
|---|---|
| Zero-shot | You are in the middle of a room. You can assume that the room's width and height are both 500 units. The layout of the room in the following format:
'name': 'bedroom', 'width': 500, 'height': 500, 'directions': 'north': [0, 1], 'south': [0, -1], 'east': [1, 0], 'west': [-1, 0], 'objects': ['name': 'chair', 'direction': 'east', 'name': 'wardrobe', 'direction': 'north', 'name': 'desk', 'direction': 'south']
Please provide the coordinates of objects whose positions are described using cardinal directions, under a conventional 2D coordinate system using the following format:
['name': 'chair', 'x': '?', 'y': '?', 'name': 'wardrobe', 'x': '?', 'y': '?', 'name': 'desk', 'x': '?', 'y': '?'] |
| Few-shot IO w/ MF | You are an expert programmer. You are in the middle of a room. You can assume that the room's width and height are both 500 units. The layout of the room in the following format:
'name': 'laundry room', 'width': 500, 'height': 500, 'directions': 'north': [0, 1], 'south': [0, -1], 'east': [1, 0], 'west': [-1, 0], 'objects': ['name': 'dryer', 'direction': 'east', 'name': 'sink', 'direction': 'west', 'name': 'washing machine', 'direction': 'south']
Please provide the coordinates of objects whose positions are described using cardinal directions, under a conventional 2D coordinate system. For example, the coordinates of objects in the above example is:
['name': 'dryer', 'x': 500, 'y': 250, 'name': 'sink', 'x': 0, 'y': 250, 'name': 'washing machine', 'x': 250, 'y': 0]
Following the examples, please give the coordinates of objects in the following room using the same format:
'name': 'bedroom', 'width': 500, 'height': 500, 'directions': 'north': [0, 1], 'south': [0, -1], 'east': [1, 0], 'west': [-1, 0], 'objects': ['name': 'chair', 'direction': 'east', 'name': 'wardrobe', 'direction': 'north', 'name': 'desk', 'direction': 'south'] |
| Few-shot IO w/o MF | You are in the middle of a room. You can assume that the room's width and height are both 500 units. The layout of the room in the following format:
'name': 'laundry room', 'width': 500, 'height': 500, 'objects': ['name': 'dryer', 'direction': 'east', 'name': 'sink', 'direction': 'west', 'name': 'washing machine', 'direction': 'south']
Please provide the coordinates of objects whose positions are described using cardinal directions, under a conventional 2D coordinate system. For example, the coordinates of objects in the above example is:
['name': 'dryer', 'x': 500, 'y': 250, 'name': 'sink', 'x': 0, 'y': 250, 'name': 'washing machine', 'x': 250, 'y': 0]
Following the examples, please give the coordinates of objects in the following room using the same format:
'name': 'bedroom', 'width': 500, 'height': 500, 'objects': ['name': 'chair', 'direction': 'east', 'name': 'wardrobe', 'direction': 'north', 'name': 'desk', 'direction': 'south'] |
| SolverLearner | You are an expert programmer. You are in the middle of a room. You can assume that the room's width and height are both 500 units. The layout of the room in the following format: 'name': 'laundry room', 'width': 500, 'height': 500, 'objects': ['name': 'dryer', 'direction': 'east', 'name': 'sink', 'direction': 'west', 'name': 'washing machine', 'direction': 'south']
Please provide the coordinates of objects whose positions are described using cardinal directions, under a conventional 2D coordinate system. For example, the coordinates of objects in the above example is:
['name': 'dryer', 'x': 500, 'y': 250, 'name': 'sink', 'x': 0, 'y': 250, 'name': 'washing machine', 'x': 250, 'y': 0]
Please summarize the pattern and implement a solver() function to achieve the goal.
def solver():
# Let's write a Python program step by step
# the input is the layout of the room
# the output the coordinates of objects
After defining the solver() function. Place the function solver() between "START_CODE" and "END_CODE". |

Table 4: Prompts for the Cipher Decryption Task.

| Mode | Prompt |
|---|---|
| Zero-shot | As an expert cryptographer and programmer, your task involves reordering the character sequence according to the alphabetical order to decrypt secret messages. Please decode the following sequence:
spring
Please answer the question by placing the decoded sequence between "START_DECODING" and "END_DECODING". |
| Few-shot IO w/ MF | As an expert cryptographer and programmer, your task involves reordering the character sequence according to the alphabetical order to decrypt secret messages. For example, given the sequence "family," you must translate it into "afilmy." Below are further examples that demonstrate the translation:
school -> chloos
Following the examples, please decode the following sequence:
spring
Please answer the question by placing the decoded sequence between "START_DECODING" and "END_DECODING". |
| Few-shot IO w/o MF | As an expert cryptographer and programmer, your task involves deciphering secret messages. For example, given the sequence "family," you must translate it into "afilmy." Below are further examples that demonstrate the translation:
school -> chloos
Following the examples, please decode the following sequence:
spring
Please answer the question by placing the decoded sequence between "START_DECODING" and "END_DECODING". |
| SolverLearner | As an expert cryptographer and programmer, your task involves deciphering secret messages. For example, given the sequence "family," you must translate it into "afilmy." Below are further examples that demonstrate the translation:
school -> chloos
Please deduce the encryption system and develop a solver() function for the decryption.
def solver():
# Let's write a Python program step by step
# the input is the coded sequence
# the output is the decoded sequence
After defining the solver() function. Place the function solver() between "START_CODE" and "END_CODE". |

Table 5: Full Main Results for Arithmetic Task.

| Method | Base | 8 | 9 | 10 | 11 | 16 |
|---|---|---|---|---|---|---|
| GPT-3.5 | Zero-shot | 0.330 | 0.117 | **1** | 0.066 | 0.294 |
| | 8-IO w/ MF | 0.376 | 0.089 | **1** | 0.089 | 0.849 |
| | 8-IO w/o MF | 0.120 | 0.027 | 0.905 | 0.057 | 0.587 |
| | 16-IO w/ MF | 0.428 | 0.088 | **1** | **0.098** | 0.912 |
| | 16-IO w/o MF | 0.108 | 0.025 | 0.924 | 0.063 | 0.575 |
| | 8-shot SolverLearner | **0.571** | **0.462** | **1** | 0.095 | **1** |
| GPT-4 | Zero-shot | 0.600 | 0.697 | 0.999 | 0.551 | 0.819 |
| | 8-IO w/ MF | 0.576 | 0.717 | 0.860 | 0.540 | 0.862 |
| | 8-IO w/o MF | 0.255 | 0.268 | 0.545 | 0.264 | 0.431 |
| | 16-IO w/ MF | 0.543 | 0.720 | 0.817 | 0.534 | 0.840 |
| | 16-IO w/o MF | 0.257 | 0.245 | 0.505 | 0.237 | 0.435 |
| | 8-shot SolverLearner | **1** | **1** | **1** | **1** | **1** |

Table 6: Full Main Results for Basic Syntactic Reasoning.

| Method / Word Order | | OSV | OVS | SOV | VOS | VSO |
|---|---|---|---|---|---|---|
| GPT-3.5 | Zero-shot | 0.560 | 0.298 | 0.190 | 0.226 | 0.560 |
| | 8-IO w/ MF | 1 | 0.643 | 0.583 | 0.976 | 0.988 |
| | 8-IO w/o MF | 1 | 0.452 | 0.929 | 0.988 | 1 |
| | 16-IO w/ MF | 1 | 0.738 | 0.762 | 0.988 | 0.952 |
| | 16-IO w/o MF | 1 | 0.190 | 0.964 | 1 | 1 |
| | 8-shot SolverLearner | 0.988 | **1** | **1** | **1** | **1** |
| GPT-4 | Zero-shot | 1 | 1 | 1 | 1 | 1 |
| | 8-IO w/ MF | 1 | 1 | 1 | 1 | 1 |
| | 8-IO w/o MF | 1 | 1 | 1 | 1 | 1 |
| | 16-IO w/ MF | 1 | 1 | 1 | 1 | 1 |
| | 16-IO w/o MF | 1 | 0.988 | 1 | 1 | 1 |
| | 8-shot SolverLearner | **1** | **1** | **1** | **1** | **1** |

Table 7: Full Main Results for Spatial Reasoning.

| Method / Coordinates | | Default | S-NS | S-WE | R90 | R180 | R270 | Random |
|---|---|---|---|---|---|---|---|---|
| GPT-3.5 | Zero-shot | 0.273 | 0.702 | 0.143 | 0.012 | 0.310 | 0.060 | 0.024 |
| | 8-IO w/ MF | 0.952 | 0.845 | 0.869 | 0.25 | 0.976 | 0.060 | 0.095 |
| | 8-IO w/o MF | 0.369 | 0.726 | 0.310 | 0.083 | 0.690 | 0.107 | 0.071 |
| | 16-IO w/ MF | 0.929 | 0.893 | 0.857 | 0.274 | 0.952 | 0.071 | **0.131** |
| | 16-IO w/o MF | 0.452 | 0.667 | 0.452 | 0.083 | 0.798 | **0.131** | 0.083 |
| | 8-shot SolverLearner | **1** | **1** | 0 | 0 | **1** | 0 | 0 |
| GPT-4 | Zero-shot | 0.119 | 0.060 | 0.083 | 0.024 | 0.048 | 0.012 | 0.036 |
| | 8-IO w/ MF | **1** | **1** | 0.964 | 0.643 | 0.952 | 0.679 | 0.190 |
| | 8-IO w/o MF | **1** | 0.976 | 0.929 | 0.560 | 0.976 | 0.429 | 0.333 |
| | 16-IO w/ MF | **1** | **1** | 0.952 | 0.690 | 0.929 | 0.667 | 0.214 |
| | 16-IO w/o MF | **1** | 0.976 | 0.964 | 0.607 | 0.976 | 0.405 | 0.369 |
| | 8-shot SolverLearner | **1** | **1** | **1** | **1** | **1** | **1** | **1** |

Table 8: Full Main Results for Cipher Decryption.

| Method / Encryption System | | Alphabetically Sorting Cipher | Caesar Cipher | Morse Cipher |
|---|---|---|---|---|
| GPT-3.5 | Zero-shot | 0.560 | **0.036** | 0.512 |
| | 8-IO w/ MF | 0.595 | 0.024 | 0.464 |
| | 8-IO w/o MF | 0.560 | 0 | 0.452 |
| | 16-IO w/ MF | 0.619 | 0.024 | 0.536 |
| | 16-IO w/o MF | 0.512 | 0.012 | 0.440 |
| | 8-shot SolverLearner | **1** | 0 | **1** |
| GPT-4 | Zero-shot | 0.726 | 0 | 1 |
| | 8-IO w/ MF | 0.774 | 0.060 | 1 |
| | 8-IO w/o MF | 0.75 | 0.583 | 1 |
| | 16-IO w/ MF | 0.798 | 0.179 | 1 |
| | 16-IO w/o MF | 0.738 | 0.583 | 1 |
| | 8-shot SolverLearner | **1** | **1** | **1** |

Table 9: Results over Claude3 for Arithmetic Task.

| Method / Base | 8 | 9 | 10 | 11 | 16 |
|---|---|---|---|---|---|
| Zero-shot | 0.710 | 0.185 | 0.996 | 0.334 | 0.868 |
| 8-IO w/ MF | **0.783** | **0.385** | 0.995 | **0.473** | 0.913 |
| 8-IO w/o MF | 0.269 | 0.083 | 0.659 | 0.105 | 0.752 |
| 8-shot SolverLearner | 0 | 0 | **1** | 0.095 | **1** |

Table 10: Results over Claude3 for Basic Syntactic Reasoning.

| Method / Word Order | OSV | OVS | SOV | VOS | VSO |
|---|---|---|---|---|---|
| Zero-shot | 1 | 1 | 1 | 1 | 0.988 |
| 8-IO w/ MF | 1 | 1 | 1 | 1 | 1 |
| 8-IO w/o MF | 1 | 0.976 | 1 | 1 | 1 |
| 8-shot SolverLearner | 1 | 1 | 1 | 1 | 1 |

Table 11: Results over Claude3 for Spatial Reasoning.

| Method / Coordinates | Default | R90 | R180 | R270 | S-NS | S-WE | Random |
|---|---|---|---|---|---|---|---|
| Zero-shot | 0.607 | 0.012 | 0.119 | 0.024 | 0.321 | 0.262 | 0.060 |
| 8-IO w/ MF | 1 | 1 | 1 | 1 | 0.988 | 0.988 | 1 |
| 8-IO w/o MF | 1 | 1 | 1 | 1 | 1 | 1 | 1 |
| 8-shot SolverLearner | 1 | 1 | 1 | 1 | 1 | 1 | 1 |

Table 12: Results over Claude3 for Cipher Decryption.

| Method / Encryption System | Alphabetically Sorting Cipher | Caesar Cipher | Morse Cipher |
|---|---|---|---|
| Zero-shot | 0.560 | 0.024 | 0.988 |
| 8-IO w/ MF | 0.607 | 0.167 | 1 |
| 8-IO w/o MF | 0.214 | 0.048 | 1 |
| 8-shot SolverLearner | 0.131 | 0.119 | 1 |

Table 13: Results over the arithmetic task with Python interpreter as executor vs. GPT-3.5 as executor

| Executor / Base | 8 | 9 | 10 | 11 | 16 |
|---|---|---|---|---|---|
| Python Interpreter | 1 | 1 | 1 | 1 | 1 |
| GPT-3.5 | 0.398 | 0.196 | 0.934 | 0.152 | 0.64 |

Table 14: Results for the spatial reasoning over GPT-3.5 w.t.r the number of few-shot examples

| Shot / Coordinates | Default | S-NS | S-WE | R90 | R180 | R270 | Random |
|---|---|---|---|---|---|---|---|
| 1 | 1 | 1 | 0 | 0 | 0 | 0 | 0 |
| 2 | 1 | 1 | 0 | 0 | 1 | 0 | 0 |
| 4 | 1 | 1 | 0 | 0 | 1 | 0 | 0 |
| 8 | 1 | 1 | 0 | 0 | 1 | 0 | 0 |
| 16 | 1 | 1 | 0 | 0 | 1 | 0 | 0 |

