# OpenReview forum: "Inductive or Deductive? Rethinking the Fundamental Reasoning Abilities of LLMs"
_ICLR.cc/2025/Conference — Submitted to ICLR 2025_

### Official Review · Reviewer_sVwR · 2024-10-27

**Soundness:** 2
**Presentation:** 3
**Contribution:** 1
**Rating:** 3
**Confidence:** 4

**Summary:**

The paper evaluates the inductive and deductive reasoning capabilities of large language models using a framework named SolverLearner. Through tasks including arithmetic, syntactic reasoning, spatial reasoning, and cipher decryption, the authors claim that LLMs perform well in inductive reasoning but struggle with deductive reasoning tasks. The topic of reasoning ability of LLMs is interesting and important. The overall presentation of the paper is clear and fluent.

**Strengths:**

(1) The presentation is clear, fluent, and easy to follow. The task formulation is clear, and the introduction of the proposed framework is transparent.

(2) The topic of reasoning ability of LLMs is crucial and in need of exploration.

(3) The experiments are detailed introduced, with all settings and prompts attached in the appendix.

**Weaknesses:**

(1) The task setup does not adequately reflect real-world scenarios. The tasks used in the evaluation, particularly the arithmetic tasks across different number bases and synthetic syntactic tasks, are highly artificial and contrived. These tasks do not reflect realistic reasoning challenges that LLMs would face in natural language processing or human cognition. For example, arithmetic problems in base-9 or base-11 are not commonly encountered in real-world settings. The syntactic reasoning tasks are also simplistic, relying on predefined sentence structures with fixed subject-verb-object patterns. More realistic scenarios, such as understanding context-dependent syntactic reordering or handling ambiguous language, would make the evaluation more robust and relevant to practical applications.

(2) The proposed framework has limited generalizability. The proposed SolverLearner, while effective for some inductive tasks, does not generalize well to broader inductive reasoning challenges. The tasks in the study are highly structured and constrained (e.g., learning the mapping function in base-specific arithmetic), where a unique solution exists for the inductive task. In more complex scenarios, such as reasoning about abstract concepts, learning open-ended rules, or inducing general principles from noisy data, the SolverLearner framework may not be effective. The paper does not discuss how this method could scale to such more complex inductive challenges, where the learning task is not well-defined and may involve multiple plausible solutions.

(3) Comparison with other reasoning frameworks is insufficient. The paper fails to adequately compare its results with alternative approaches to reasoning in LLMs, such as chain-of-thought prompting, least-to-most prompting, or retrieval-augmented generation. While SolverLearner is presented as a novel method for isolating inductive reasoning, the lack of comparison with existing techniques leaves its relative merits unclear. For example, chain-of-thought prompting has been shown to improve both inductive and deductive reasoning in various tasks by breaking complex problems into smaller reasoning steps. Without a direct comparison, it is difficult to assess whether SolverLearner offers any significant advantage over these established methods. Including such comparisons would have strengthened the evaluation.

(4) The scope of deductive evaluation is too narrow. The deductive reasoning tasks primarily focus on counterfactual arithmetic (e.g., base-9 vs. base-10 arithmetic), which is a very specific case. Deductive reasoning encompasses more than just counterfactual logic—it includes formal logic, rule-based reasoning, and mathematical proofs. The paper does not evaluate these broader aspects of deductive reasoning, such as tasks that involve symbolic logic, proof generation, or formal theorem proving. This limited scope weakens the claim that LLMs perform poorly in deductive reasoning overall. For example, the study might have included tasks like syllogisms or multi-step logical deductions, which would provide a broader view of LLMs' deductive reasoning capabilities.

**Questions:**

The paper claims that SolverLearner isolates inductive reasoning, but this separation is not convincingly demonstrated. For example, in the arithmetic task of base-8 addition, the process of identifying the base from examples is considered as inductive reasoning. I wonder if the authors could provide more convincing evidence to show that the model is truly performing inductive reasoning instead of simply pattern matching based on prior exposure to similar tasks?

Note that, using Python interpreters to prevent LLM involvement in the "deductive" step (function execution) does not fully eliminate the possibility that LLMs leverage both types of reasoning in the previous “inductive" step. The distinction remains unclear because the task structure could involve deductive elements when identifying input-output mappings.

---

### Official Review · Reviewer_ee9E · 2024-11-03

**Soundness:** 2
**Presentation:** 2
**Contribution:** 2
**Rating:** 5
**Confidence:** 3

**Summary:**

The proposed approach aims to disentangle deductive from inductive capabilities of an LLM. The main contribution is a series of tasks where each task is has both an inductive as well as a corresponding deductive component. The results show that LLMs perform more poorly in deductive reasoning as compared to inductive reasoning on the tasks designed to test both.

**Strengths:**

Strengths
- Disentangling the inductive and deductive capabilities of LLMs seems like an interesting problem
- The types of benchmarks used are varied and several of the state-of-the-art LLMs have been considered in the evaluation

**Weaknesses:**

Weakness
- The paper seems to suggest that solverlearner is a novel approach, but it is less clear why this is the case. As far I could understand, solver learner just utilizes an external code interpreter to apply the functions learned by the LLM inductively. I was not clear of the complexity involved to do this, since the approach itself is not described in detail. Further, are there other ways of decoupling the two, since there was not a lot of context in why this is the right way for disentanglement.
- The tasks itself also seem to be from prior work (Wu et. Al 2023) apart from the cipher task. Once again, I was not sure if the contribution of the tasks was significantly different from prior work.
- Regarding the foundational aspect as such, based on the definition of deductive/inductive inference, since the LLMs are being used a bit like black-boxes, I was not sure about the leap from observing the experimental results to concluding the “type” of inference the LLM is truly performing internally. For e.g. memorization is one aspect that could be affecting the way a LLM is solving a particular task.
- In terms of the significance of the study, is the fact that deduction is harder than induction significant, i.e., what would be a good application use-case to motivate this study is something that was missing.

**Questions:**

- Some additional comments about the novelty of the proposed evaluation and its significance in LLMs would be useful.

---

### Official Review · Reviewer_Nc7T · 2024-11-03

**Soundness:** 3
**Presentation:** 4
**Contribution:** 3
**Rating:** 5
**Confidence:** 3

**Summary:**

The paper starts from the observation that there are two forms of   reasonung in LLMs: inductive and deductive. The authors empirically study prformance of the two approaches, and conclude that LLMs do better with induction.

**Strengths:**

The authors present several contributions to a major research domain in IA:
They introduce a notion of several forms of reasoning going on the system,
They implement a system to validate their claims.
They experimentally obtain unexpected results

The paper is also rather easy to follow.

**Weaknesses:**

I have two major difficulties with the paper:

As the authors themselve observed, the two forms of reasoning are very much intertwined. ALthough Fig 1 does a nice job at explaining the concepts,  as i read I felt the need for a more formal definition of what is induction and what is deduction. This is especially true when I looked at the discussion, and I felt I could not understand why statements such as
". By
completely disentangling the inductive reasoning of LLMs, our proposed SolverLearner shows the
remarkable inductive reasoning capabilities inherent in LLMs."

I also felt the notions of induction and deduction may take somewhat different meanings for different researchers

Second, I would have hoped for a deeper insight into these results. You mention the remarkable induct reasoning of LLMs, but it would be nice (at least for me) to understand how they appear. Also, why deduction performs worse?

Function execution: you state it takes place outside the LLM (eg, in Python). Why are the differences so big (Table 13?)

**Questions:**

Other problems/questions:

I understand why you use chat-gpt, but it does make your work non-reproducible.  It would be helpful to complement results with an open-source system (it would also help in making your conclusions more general).

Function Execution: if you do it outside the LLM, is it still a LLM?

8-IO w/ Mapping Function (MF): is this deductive or a mix?

The results show a noticeable improvement between chat-gpt 3.5 and 4. ANy idea why?

There are a few typos and in a few cases bad english "

Wu23 Paper: why only datasets from this work. It seems highly related, why is it not discussed?

---

### Official Review · Reviewer_Rgex · 2024-11-04

**Soundness:** 3
**Presentation:** 2
**Contribution:** 2
**Rating:** 5
**Confidence:** 4

**Summary:**

This paper studies LLM capabilities in inductive and deductive reasoning, and compares the performance gap between the two poles of reasoning. They consider this by framing reasoning as a function (which connects input and output) definition task, with
- deductive: the model is provided with the function (direct input-output mappings)
- inductive: the model is given examples (x,y) pairs but without the function

With the framework defined, they test the reasoning processes of LLMs across 4 primary subtasks: arithmetic, basic syntax reasoning (syntactical recognition and identification), spatial reasoning, and a novel cipher decryption task of their own design. Their finding suggests that LLMs seem to be stronger inductive reasoners rather than deductive. In particular, tasks that involve counterfactual reasoning are particularly challenging even with strong inductive performance.

**Strengths:**

I think the main strength of this task is that it provides a new way of looking at various ways we interact with LLMs. Using a framework of inductive and deductive reasoning, we can perhaps consider in-context learning as a deductive reasoning task and code generation as an inductive reasoning task. I think it would have been more salient to frame the paper this way, and therefore, more relevant to many other communities, particularly with LLM evaluation communities. Below describes the strengths in more specifics.

- S1. This paper discusses the distinction between inductive and deductive reasoning, and how we may systematically investigate this using a novel framework.
- S2. In providing a novel framework, they provide a novel task of their own design. A cypher description task . This is a particular strength, because many available evaluative framework and benchmarks could have already been used in pretraining of many closed LLMs. With an introduction of a novel task, they can robustly test.
- S3. This potentially adds a novel way of looking at code generation and reasoning together.
- S4. I can see how we use in-context learning could be considered from the perspective of inductive and deductive reasoning with this framework.
- S5. The framework effectively describes a spectrum between deductive and inductive reasoning. Inductive and deductive reasoning are not always distinctly delineated as I had previously conceptualized, so it was interesting for me to consider.
- S6. The paper was well written with mostly clear description.
- S7. This framework and subtasks were thoroughly experimented with many current SOTA LLMs

**Weaknesses:**

As pointed out in the strengths, I think this paper has the potential to make us consider looking at how we interact with LLMs in a novel way. However, I think the formulation of the question and presentation of the main thesis could be significantly improved. The paper does not adequately situate itself with existing literature on reasoning, or discuss the relations between this work with respect to code generation or in-context learning.

- W1. The paper and their findings were hard to connect to existing work. I think it would have made the paper stronger to consider how other work in reasoning area compare with this approach. There have been numerous work on deductive, inductive, abductive, counterfactual etc reasoning. I think there was very few discussion on the prior work, and therefore, this work was poorly situated.

- W2. The framework relies on a particular case of generation, which is code generation. I think the performances could very much differ in the case of natural language generation and inductive reasoning. I think it's insufficient to generalize the findings of this paper to the broad deductive/inductive reasoning gap of LLMs

- W3. "Current methods that investigate deductive and inductive reasoning often rely on disparate datasets" may not true: LogiGLUE (https://arxiv.org/pdf/2310.00836) for example considers both categories in their datasets.

- W4. I believe there were prior work on inductive and deductive reasoning, and some of these prior work does discuss the gap between them. On the claim of novelty, I believe that the question itself is not novel enough. The framework may be novel.

- W5. This work does not consider finetuned models, but it would have been interesting to consider them, particularly in discussion with deductive/inductive reasoning and seen examples. The paper does mention some probable explanation for some performance gaps on examples seen/unseen during pretraining.

- W6. I think there could have been discussion on how this relates to the code generation performance. There are many existing benchmarks for code generation (e.g. BigCodeBench, HumanEval), and because this framework relies on code generation for an external executor, it should be discussed as what these evaluative benchmarks and testing on with respect to deductive/inductive reasoning.

- W7 The writing could be improved in some parts of the paper. I found the deductive part to be a bit lacking in discussion.


I believe this work has a lot of potential and it was very interesting to read about this framework! I hope to see this work out, but I wish it was more thoroughly considered and better presented/situated in connection with existing work in the field.

**Questions:**

- Q1. Has inductive and deductive reasoning not studied before in previous literature? I believe there are existing work or at least work on either of the categories. (for instance, for deductive reasoning, see: https://aclanthology.org/2023.findings-acl.67.pdf, https://openreview.net/forum?id=KFjCFxiGk4; for inductive reasoning, see: https://arxiv.org/abs/2309.05660). Please review existing literature and include them in your related work. Perhaps the distinction between the two has not been made explicit, which I believe is a fair contribution, but please acknowledge existing work.
- Q2. How do you expect this to connect to benchmarking and evaluations of LLMs? How would this improve robustness of LLMs?
- Q3. I think this work could benefit by considering the tension between memorization (as briefly discussed in the paper about models performing better on the examples seen during the pretraing phase) vs. in-context learning. What would be the connection of inductive/deductive reasoning and in-context examples in this framework?
- Q4. Why is it important to distinguish deductive and inductive reasoning? (I believe it *is* important, but I wish the authors to consider this question. In my opinion, it could be useful particularly in the application of LLMs and improving performances of various symbolic reasoning involved generation such as code generation, scientific LLMs, or verification/formal language modeling with LLMs. Perhaps if the work was situated better and considered within the context of generation problems, the motivation behind this distinction would have been better argued in the paper.)
- Q5. Do you intend to release the datasets and prompts used for the tasks?

---

### Meta-Review · Area_Chair_fJ57 · 2024-12-18

**Metareview:**

All reviewers agreed that this paper potentially provides a novel way to understand the interaction with LLMs. The set of benchmarks is varied, and the paper is rather well-written.

However, the novelty should be made clearer, and the work should provide a better placement within the existing literature on reasoning, in particular with respect to code generation or in-context learning. Moreover, a clearer and more formal definition of what is induction and what is deduction in this context is needed. Real-world scenarios or at least the potential to aim for those should be discussed better.

**Additional Comments On Reviewer Discussion:**

Since there was no rebuttal provided by the authors, the listed weaknesses of the paper could not be resolved in the discussion.

---

### Decision · Program_Chairs · 2025-01-22

Reject